# Composite Single Lap Shear Joint Integrity Monitoring via Embedded Electromechanical Impedance Sensors

Steven P. Caldwell 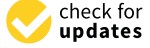 and Donald W. Radford *

Composite Materials, Manufacture and Structures Laboratory, Colorado State University, Fort Collins, CO 80523, USA
* Correspondence: donald.radford@colostate.edu

**Abstract:** Composite bonded structure is a prevalent portion of today's aircraft structure. Adequate bond integrity is a critical aspect of fabrication and service, especially since many of today's structural bonds are critical for flight safety. Over the last decade, non-destructive bond evaluation techniques have improved but still cannot detect a structurally weak bond that exhibits full adherend/adhesive contact. The result is that expensive and time-consuming structural proof testing continues to be required to verify bond integrity. The objective of this work is to investigate the feasibility of bondline integrity monitoring using piezoelectric sensors, embedded at different locations within the composite joint, and to assess the benefits of monitoring the thickness mode in addition to the radial mode. Experiments and analyses are performed on single lap shear composite joints, with and without embedded sensors, subjected to incrementally increasing tensile loads. The results indicate that the embedded piezoelectric sensors measure a change in the resonance in both the radial and thickness mode during incremental loading and that the thickness resonance shows enhanced sensitivity to impending failure. Thus, it is demonstrated that monitoring both modes of the piezoelectric sensor provides addition details for prognostic performance evaluation.

**Keywords:** structural health monitoring; single lap shear joint; piezoelectric sensor; composite bond; electromechanical impedance; finite element modeling



## 1. Introduction

Joints connected with mechanical fasteners worked well with metallic aircraft structures of the past. With the advent of lightweight composite aircraft structure, the use of bolted joints has become inefficient both due to the fastener hole drilling process and to the reduced shear-out resistance of composites versus metal counterparts. The use of adhesively bonded joints in composite designs provides the characteristics of efficient production, light weight structure, and ease of maintainability [1–3].

An effective adhesive joint relies on correctly designed laminated composite adherends, an appropriate adhesive, and a strong interface between the adherend and the adhesive. While designing the composite adherend and choosing an appropriate adhesive are relatively well understood and readily controlled, a major downside is that the joint integrity is extremely sensitive to process controls during the bonding operation. Poor process control during bonding can result in reduced bond interfacial strength and lead to a structurally weak bond that traditional non-destructive inspection techniques cannot detect since the adherends remain in full physical contact [4–6]. These types of weak bonds are sometimes referred to as "kissing" bonds and can exhibit a zero-volume disbond. These weak bonds could be due to many factors including improper bond surface preparation, surface contaminates, adhesive curing issues, and residual stresses. Bond defects of this nature can appear locally during fabrication and potentially grow during service due to repeated loadings and moisture ingression, resulting in a compromised joint. Thus, there is a need for initial bond monitoring and future evaluation of the bonded joint structural health.

Health monitoring of composite structures is relatively new, but research has increased dramatically over the last ten years [7–11]. These studies offer approaches to evaluating the health of bonded composite assemblies, immediately after manufacture and throughout the service life of the aircraft. Most of the research into structural health monitoring of composites is focused on damage to the composite material itself. Sensors are typically placed on the surface of the composite and sense damage to the external surface. Composite material failures are different from failures in metallic materials and occur by very different mechanisms. Composites fail differently in tensile loading than they do in compression loading. They are prone to hidden damage from low velocity surface impacts, which is referred to as barely visible damage. Fastener holes in composite material require more complex analyses than those in metallics due to the anisotropic nature of the composites.

Currently, in the industry, there are many types of sensors that may be attached to the structure for health monitoring. They range from traditional strain gages and accelerometers to fiber-optic strain sensors and piezoelectric ceramic sensors. Most of these sensors operate in both the static and dynamic regimes. To note a local change in stiffness, the sensor must be sensitive to small changes in the response of the structure to excitation. The traditional external surface-mounted sensors may not have the sensitivity to be able to measure a local degradation in the bondline integrity under static and dynamic loadings. Another option is to embed a sensor in the adhesive bondline offering a potential method to directly monitor the structural health state of the bonded joint. However, concerns exist about (i) the potential reduction in joint performance due to the inclusion of the sensor itself, and (ii) the location of the sensor within the joint.

Much work has been focused on piezoelectric sensors [12–15]. These sensors are small, lightweight, relatively low in cost, and can be used in a passive or active mode. In the active mode, they work as both actuators and sensors, thus providing efficient means to excite and measure the response with a single device. The use of these sensors has evolved from surface-mounted sensing to embedded bondline sensing for bonded joint health monitoring. The initial application of these sensors was used for damage detection from impacts resulting in cracks and degraded structural integrity [16]. These are proven to be sensitive enough to detect barely visible damage, making them powerful health monitoring sensors.

In recent research, for a composite structure joined with adhesives, small, piezoelectric sensors have been inserted in the bondline to evaluate joint integrity. These sensors monitor the local electromechanical impedance signature of the joint in a simple procedure that excites and records the response of the in situ sensor [17,18]. These sensors are transducers, based on the piezoelectric principle, that couple the electrical and mechanical properties. The piezoelectric effect occurs as a voltage is applied, creating an electric field that causes the sensor to lengthen or shorten according to the field polarity and proportional to the strength of the field. This piezoelectric coupling exists in both the in-plane strain (radial) direction and the out-of-plane strain (thickness) direction and can therefore can simultaneously produce a response for loads occurring on these axes. The sensors are very sensitive to minute changes in impedance occurring at high frequencies, well above the primary joint structural resonance. The minute changes serve as health indicators for the joint integrity. This capability can prove to be beneficial for an aircraft joint that is exposed to loads in multiple directions. For this application, they serve as embedded modal sensors, and the electromechanical impedance method is used to characterize the structural integrity of the bonded joint. In place, the sensor acts as both a transmitter and a receiver, functioning in a pulse-echo mode. A tone-burst voltage signal applied to the sensor generates a wave through the structure. The sensor receives the wave and passes it to the impedance analyzer, which performs a fast Fourier transform (FFT) and plots the wave resonance in the frequency domain. The strength of the returned wave provides a mechanism to evaluate joint integrity through the impedance measurement. The impedance is a complex quantity, and the real part reflects clearly defined resonances that are indicative of the coupled dynamics between the PZT sensors and the frequency-

dependent pointwise structural stiffness of the sensor in the composite joint. The real part of the electromechanical impedance measures the pointwise mechanical impedance of the structure, and the impedance spectrum is equivalent to the pointwise frequency response of the structure. As damage develops in the structure, in the vicinity of the sensor, the pointwise impedance changes. Furthermore, the stress state in the joint can vary with position. Therefore, piezoelectric sensors must be placed at critical locations in the structure, where they have high sensitivity to impending damage, but also where they have the least negative effect on the mechanical performance of the joint. The integrity of the sensor itself is independently confirmed using the imaginary part of the impedance. This part is highly sensitive to a disbond of the sensor in the adhesive, but much less sensitive then the real part to structural resonances [19]. The resonances that are observed in the real part of the impedance comprise in-plane and out-of-plane responses [20]. The in-plane responses directly relate to the shear and tension/compression joint capability, whereas the thickness or out-of-plane responses relate to the flexural or joint pull-off capability. The in-plane resonances are generally much lower in frequency than the out-of-plane resonances due to the larger dimension of the sensor in the radial direction when compared to the thickness direction.

The work herein focuses on ultrahigh frequency response impedance measurements of a sensor embedded in the composite joint bondline, specifically a single lap shear joint, subjected to excitation in the form of a constant voltage frequency sweep. The goal is to evaluate the joint health and the interaction of the sensor and its location within the joint bondline and determine how the joint performance and sensing capability is altered by moving the sensor within the bondline. Furthermore, this work measures and evaluates the effectiveness of both the radial and thickness impedance changes in assessing joint health.

## 2. Experimentation

Experiments were developed to demonstrate the ability of the embedded sensors to quantify the electromechanical joint impedance response during successive, incrementally increasing tensile loadings until joint failure. Two sensor locations within the lap shear joint were evaluated to better understand the relationship between the sensor effectiveness and any negative effects on joint performance. These two locations were chosen since they exhibit the minimum and maximum shear stress in the bond during tensile loading. In order to establish a baseline impedance response, each sensor was tested in a free state, measuring the response of the primary radial and thickness resonances. The sensors were catalogued and then used in the fabrication of standard composite lap shear coupons created via a secondary bond cure, where each sensor was inserted between two layers of film adhesive that formed the bond between the composite adherends. The coupons were then tested with incrementally increasing loads until failure, while measuring the sensor impedance between loads.

### 2.1. Sensor Free State Electromechanical Impedance

Several piezoelectric ceramic disk sensors from APC International were evaluated in the free boundary condition state to determine the radial and thickness electromechanical impedance resonances. The sensor disks were solid, consisting of material 851, which is highly purified lead zirconate titanate (PZT) ceramic [21]. The disks were 5.8 mm in diameter and 0.20 mm thick and were selected for their small geometric footprint in the adhesive bond. The sensors were provided with 150 mm leads, soldered to the top and bottom sensor surface for connection to the network analyzer to perform voltage frequency sweeps. One of the sensors with the leads attached is shown on Figure 1. APC performed the wire soldering, and each delivered sensor was measured for thickness, diameter, and wire thickness. Of note, the thickness of the sensors was measured consistently at 0.39 mm, almost double the specified thickness without soldered surface leads. The measured radial dimensions were at the specified 5.80 mm dimension and the red and black wire diameters were 0.50 mm.

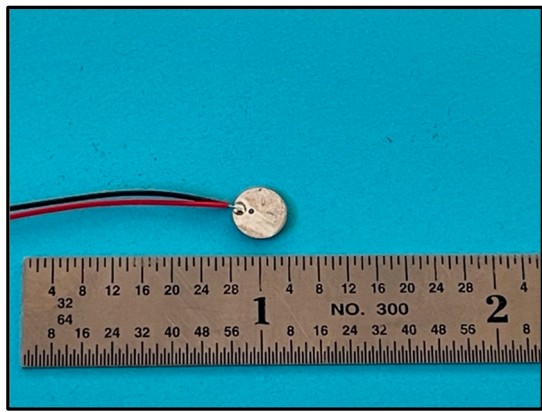

**Figure 1.** Piezoelectric sensor with leads attached.

Prior to performing the sensor free boundary state experiments, an analytical model of the sensor was created using the Ansys [22] code to simulate the impedance response. The model is shown on Figure 2 and was run in the free boundary condition state to match the experiment. The element mesh density through the thickness was initially selected at 4 elements. Note that element density along the radial and thickness axes can be increased if model calibration is required. The SOLID226 20 node element was used to represent the sensor since it includes a voltage degree of freedom for impedance analysis.

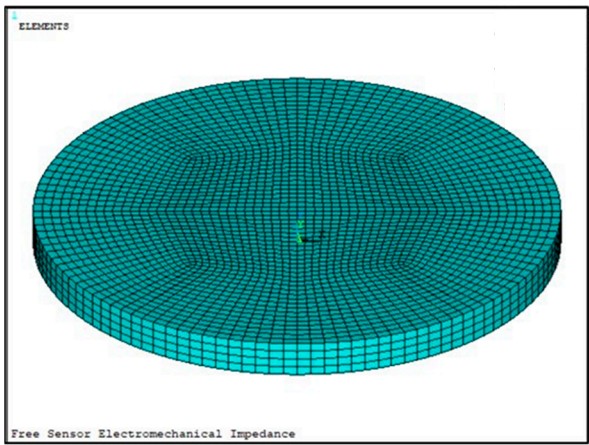

**Figure 2.** Sensor disk analytical model.

The mechanical and electrical properties representing the sensor are shown below:

$$[C_p] = \begin{bmatrix} 97 & 49 & 49 & 0 & 0 & 0 \\ 49 & 97 & 44 & 0 & 0 & 0 \\ 49 & 49 & 84 & 0 & 0 & 0 \\ 0 & 0 & 0 & 24 & 0 & 0 \\ 0 & 0 & 0 & 0 & 22 & 0 \\ 0 & 0 & 0 & 0 & 0 & 22 \end{bmatrix} \text{GPa,} \tag{1}$$

$$[\varepsilon_p] = \begin{bmatrix} 947 & 0 & 0 \\ 0 & 947 & 0 \\ 0 & 0 & 605 \end{bmatrix} \times 10^{-8} \text{ F/m,} \tag{2}$$

$$[e_p] = \begin{bmatrix} 0 & 0 & 0 & 0 & 12.84 & 0 \\ 0 & 0 & 0 & 12.84 & 0 & 0 \\ -8.02 & -8.02 & 18.31 & 0 & 0 & 0 \end{bmatrix} \text{C/m}^2, \tag{3}$$

where $[C_p]$ is the stiffness matrix, $[\varepsilon_p]$ is the dielectric matrix, and $[e_p]$ is the piezoelectric matrix [23]. The density of the sensor material is 7600 kg/m$^3$. In order to represent the experiment, a coupled field harmonic analysis is run by applying a unit voltage differential between the upper and lower disk surface and recovering the generated charge that is summed to create the induced current. The current response and the applied unit voltage level are used to compute the electromechanical impedance. The frequency analysis was run at 1 Hz intervals in the range of 200 kHz to 6000 kHz to cover the radial and thickness response resonance frequencies. The free sensor analytical impedance response (real and imaginary) is shown on Figure 3 with the primary radial and thickness resonances noted.

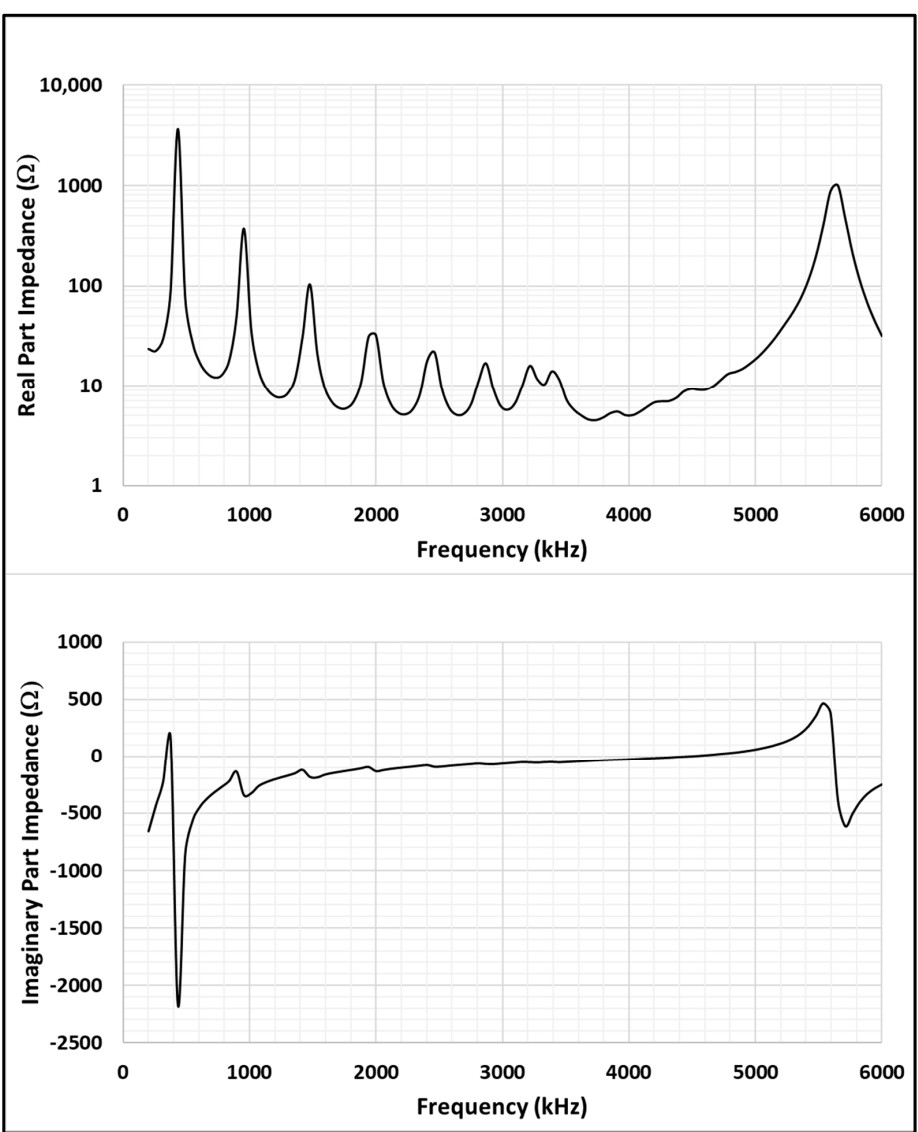

**Figure 3.** Free sensor analytical impedance response.

A total of 9 sensors were tested in the free boundary state by suspending each one by its lead wires through a rubber grommet. While this is not a totally free state in the radial direction, it offers a consistent baseline impedance response of each sensor prior to being embedded in the adhesive bondline of the test coupons. The sensor leads were connected to a Bode 100 Network Analyzer and a voltage was applied across the disk surface with one side acting as ground and the other being excited [24,25]. The voltage was applied in a linear frequency sweep from 200 kHz to 20,000 kHz. The frequency sweep takes about three minutes to complete and comprised 16,501 frequency points. For the first sensor test,

the sweep was run multiple times obtaining the same results. After that, each additional sensor was tested only a single time. For each run, the data was exported for plotting. The real part of the electromechanical impedance, for each of the 9 sensors, is plotted on Figure 4. The strongest peak is the 1st radial resonance occurring at an average of 384.5 kHz. The thickness resonance occurred at a higher frequency (5589 kHz), since the disk was thin compared to its radius.

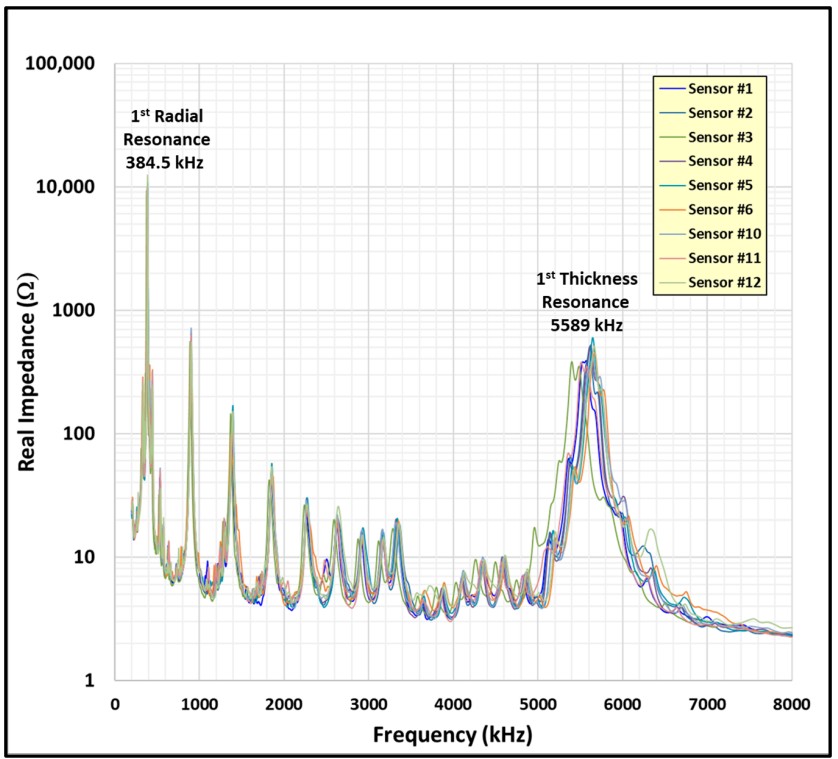

**Figure 4.** Free sensor real impedance responses.

The 9 sensors tested were remarkably close in frequency response (resonance frequency and amplitude). This consistency in response is important, because it indicates that the sensors have essentially the same impedance signature in the free state. Therefore, the results of the lap shear coupon impedance are not expected to contain any significant variances from the embedded sensor itself. Table 1 shows the primary mode resonance frequencies and amplitudes, with the averages noted at the bottom of the table.

**Table 1.** Free sensor measured impedance resonances.

| Sensor | Radial (kHz) | Amplitude (kOhms) | Thickness (kHz) | Amplitude (Ohms) |
|:---:|:---:|:---:|:---:|:---:|
| 1 | 381.2 | 7.8 | 5559 | 381.4 |
| 2 | 382.4 | 9.4 | 5604 | 490.8 |
| 3 | 375.2 | 9.2 | 5422 | 307.3 |
| 4 | 387.2 | 9.7 | 5604 | 452.9 |
| 5 | 388.4 | 10.7 | 5650 | 595.4 |
| 6 | 386.0 | 8.0 | 5650 | 398.3 |
| 10 | 388.4 | 9.2 | 5650 | 409.1 |
| 11 | 381.2 | 10.1 | 5513 | 378.1 |
| 12 | 387.2 | 12.4 | 5650 | 455.3 |
| Average | 384.5 | 9.5 | 5589 | 441.3 |

### 2.2. Single Lap Shear Coupon Design

In order to assess and quantify the adhesive bond stress distribution and the effects of the embedded sensor and its location within the bond, a finite element model representation was created and analyzed prior to testing. The finite element model with the embedded center bond sensor is shown on Figure 5.

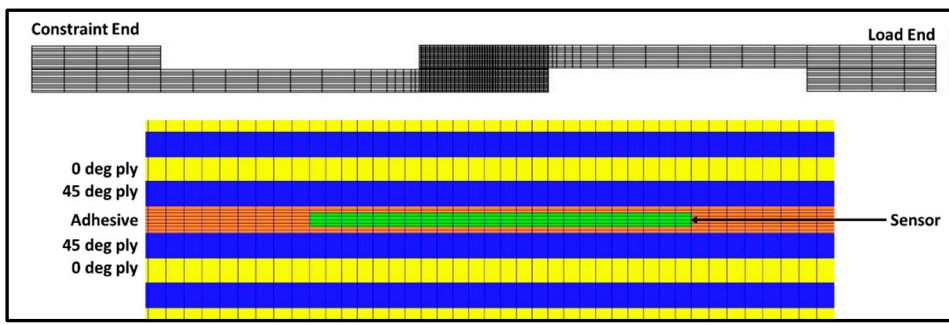

**Figure 5.** Test coupon static analysis FEM.

The sensor was directly connected to the adhesive elements at the nodal locations. The model was run statically with one end constrained and an enforced displacement applied to the free end. The sensor mechanical properties were the same as previously presented. The sensor took up approximately 4% of the shear bond area, so no significant joint degradation was expected due to a simple reduction in the bond area related to the embedded sensor. The simulation results showing the maximum shear stress distribution through the bondline are shown on Figure 6 for the pristine, center-embedded sensor, and load end-embedded sensor.

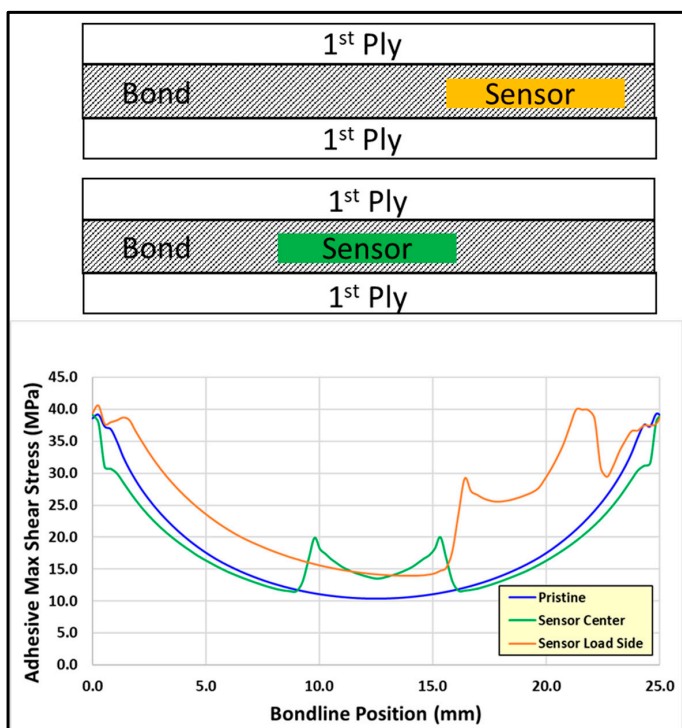

**Figure 6.** Analytical lap shear adhesive bond maximum shear stress distribution.

For the pristine case, it is seen that the maximum shear stress through the bond thickness occurs at the endpoints of the bond area. There is about a 4:1 ratio between the center and endpoint maximum stresses. The analytical results further indicate that the shear

bond performance degrades when the sensor is embedded in the area of maximum shear stress. There is a stress concentration at the edge of the bondline and adding the sensor to this location results in a higher imposed shear stress than the pristine case. Embedding the sensor at the center of the bond (the minimum stress position) results in a similar sensor effect on the stress distribution, but due to the location, the stress intensification at the sensor edges does not approach the stress seen in the pristine case at the bond endpoints. Based on these results, the bondline sensor locations for testing were selected to be the center position and the load end position. This covers the range of minimum to maximum sensor effect on the bondline shear stress distribution.

### 2.3. Adhesively Bonded Composite Single Lap Shear Coupon Fabrication

Static shear testing was performed on single lap shear test coupons with, and without, embedded piezoelectric sensors in the bondline. The coupons were fabricated with composite adherends that consisted of 12 plies of T650-35/5320-1 carbon/epoxy 8HS woven fabric prepreg in a quasi-isotropic layup $((45/0)_3)_S$. The laminates used the Solvay CY-COM 5320-1 toughened epoxy resin system and are vacuum-bag cured, out-of-autoclave, at 177 °C [26]. This resulted in an adherend thickness of 4.5 mm (0.376 mm thickness per cured ply). Figure 7 shows the layout of the adherends before the application of the adhesive.

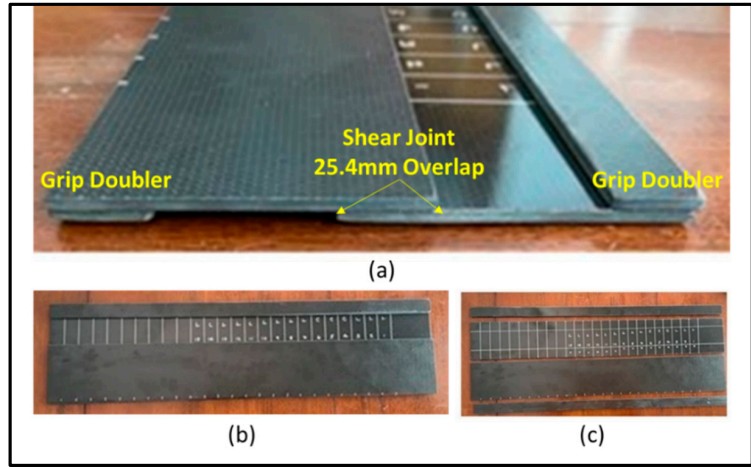

**Figure 7.** Adherend layout before adhesive cure: (**a**) side view, (**b**) plan view, (**c**) part view.

The adherends were vacuum-bagged and secondarily bonded together with two layers of FM300-2K film adhesive at 121 °C for 90 min [27]. Figure 8 shows the bonded coupons after the adhesive cure, but before cutting into individual test coupons. Individual coupon dimensions are shown on Figure 9.

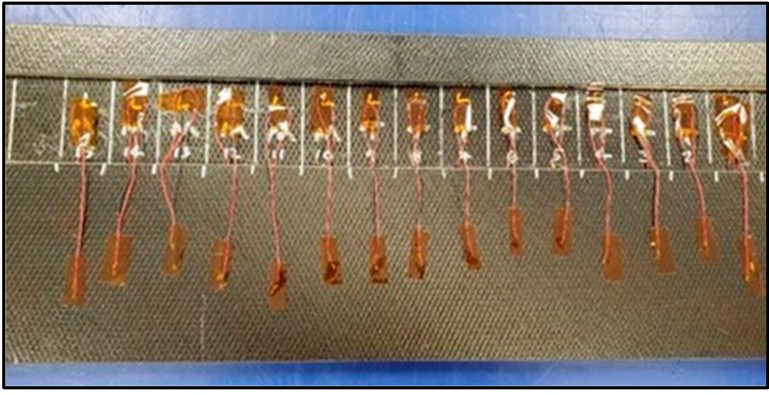

**Figure 8.** Bonded coupons after adhesive cure.

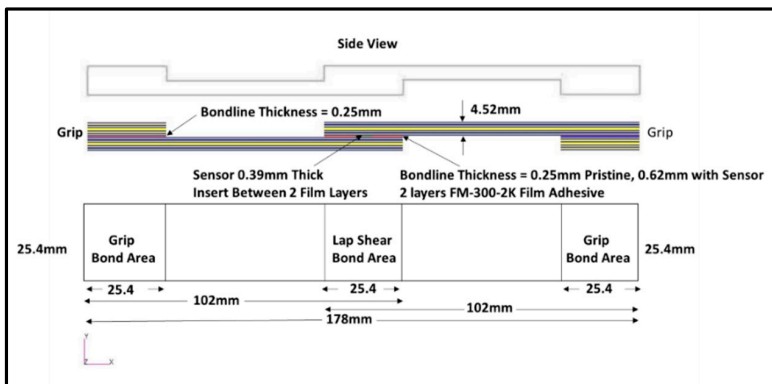

**Figure 9.** Lap shear test coupon dimensions.

A total of 5 pristine- and 9 bondline-embedded sensor lap shear test coupons were fabricated. The pristine coupons (no embedded sensor) were tested to establish a baseline failure load for comparison to the failure load with the bondline-embedded sensor. The sensors were placed between the two layers of film adhesive to encapsulate the sensor uniformly and consistently in the bond. The sensor locations within the bondline were varied, as shown in Figure 10. The locations were selected based on the simulation results and this was carried out to evaluate the sensitivity of the impedance measurement to the sensor location and to the bond shear stress distribution.

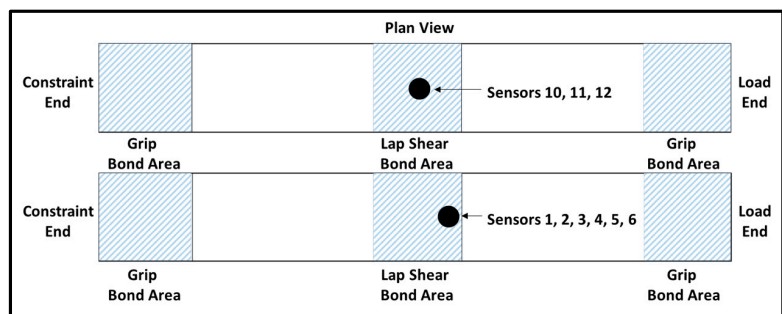

**Figure 10.** Test coupon-embedded sensor locations.

The completed test coupons are shown, in Figure 11, prior to tensile testing. Note that the silver pen-marks (circles) indicate the position of the sensor embedded in the bond and are not the actual sensors. The shear joint bondline thicknesses were measured using a micrometer, resulting in a consistent thickness of 0.62 mm.

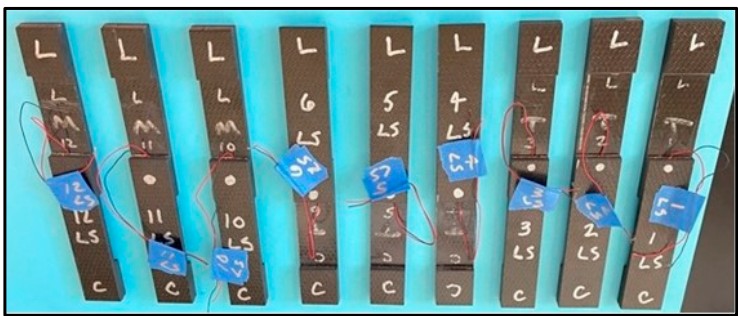

**Figure 11.** Embedded sensor lap shear test coupons.

*2.4. Coupon Tensile Load and Impedance Testing*

The ASTM D5868 FRP single lap shear test procedure was followed, loading the coupon in tension, producing shear stress in the bond [28]. The coupons were loaded at

a rate of 1.25 mm/min using the applied test system (ATS 900) universal testing machine. This is a deviation from the recommended load/deflection rate of 13 mm/min since the embedded sensors must be unloaded between each applied load to measure the impedance. Given this lower load rate, each coupon test takes about 4 min to complete each incremental static loading, allowing the operator to better control accuracy of attaining the setpoint loads. After completing each load step and returning to zero load, the coupon was removed from the test machine and the embedded sensor electromechanical impedance was measured in a freestanding, unloaded state, similar to the baseline test. Coupon impedance was measured outside the test machine in a free state due to the effect of the test machine grips on the impedance measurement. The sensors are believed to be sensitive to the characteristics of the test machine grips from both the mechanical loading and the electrical properties of the metal. The measured impedance of the coupon installed in the load machine did not correlate to the freestanding results; therefore, each coupon was removed from the test machine between incremental load applications, and its impedance was measured in the freestanding, unloaded state. For testing, each coupon is first loaded to 4448 N then unloaded for impedance measurement, followed by reinsertion in the test grips and reloading to a load value 445 N higher than the previous increment, until failure. The test procedure is shown in Table 2.

**Table 2.** Single lap shear coupon test procedure.

| Test Procedure | |
|---|---|
| **Step** | **Action** |
| 1 | Measure embedded sensor impedance. |
| 2 | Insert coupon into the universal test machine. |
| 3 | Photograph the test specimen. |
| 4 | Apply an initial load of 4448 N, unload, remove coupon and measure impedance. |
| 5 | Reinsert the coupon into the test machine and load to 4893 N (an increment of 445 N), unload, remove and measure impedance. |
| 6 | Repeat sequence adding 445 N until failure. |
| 7 | Record individual peak loads (and average). |
| 8 | Record test load/deflection and save to an Excel file. |
| 9 | Photograph broken specimens. |
| 10 | Identify failure mode type per ASTM 5573-99 [29]. |

Figure 12 shows the embedded sensor coupon, #11, in the test fixture. The clip gage extensometer was attached to the coupon to measure the shear bond axial displacement during loading. The displacement was recorded along with the applied load.

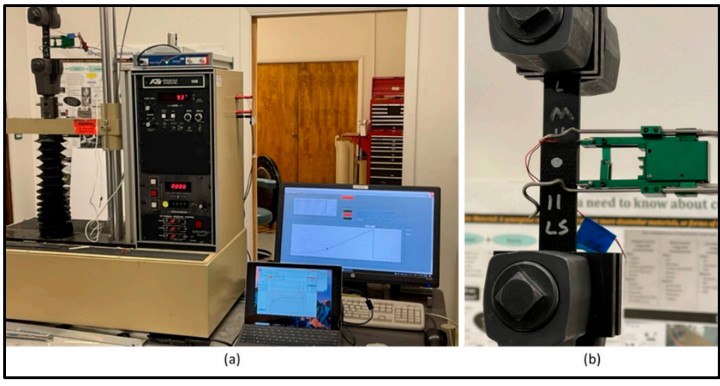

**Figure 12.** ATS 900 test system with coupon loaded. (**a**) Test setup, (**b**) coupon closeup.

## 3. Results

The free sensor impedance measurements are shown in the Experiments section along with the analytical simulation results. These are reviewed in the discussion section.

### 3.1. Pre-Test Baseline Test Coupon Electromechanical Impedance Measurements

The electromechanical impedance of each embedded sensor coupon was measured prior to lap shear testing. This established a baseline pre-load impedance and was also used to compare to the free sensor's impedance. This comparison established the difference between the sensor free state impedance and the lap joint-embedded sensor's impedance. As a result of these baseline tests, it is noted that only four of the nine embedded sensors are fully functional. The five coupons with inoperable sensors had no clear radial or thickness impedance resonances in the expected frequency range of interest. This may be due to problems occurring during the coupon bonding operation and demonstrates a need for redundant sensors if these are used in aircraft bondline integrity monitoring. Due to this result, only the four coupons with functioning sensors were tensile tested. These are coupons 2, 3, 11, and 12. Figure 13 shows a plot of the real and imaginary components of the embedded sensor impedance. These embedded sensors were averaged and show a consistent impedance response with the first radial impedance resonance at 656 kHz and the first thickness resonance at 5200 kHz. Figure 14 shows an example of one of the inoperable sensors compared to the average of the operable sensors. As seen in the figure, there are no appreciable radial or thickness resonances in the frequency range, consistent with the operable sensors, suggesting either a failure during the cure of one of the wire leads, or of the brittle piezoelectric sensor itself.

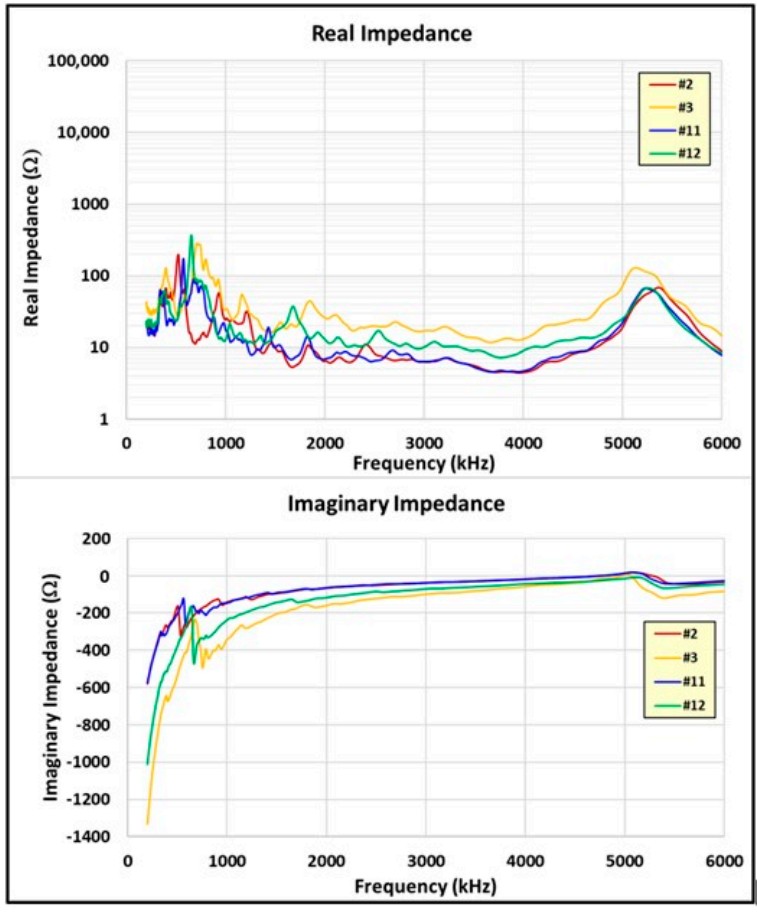

**Figure 13.** Pre-load test embedded sensor impedance resonances (Sensors 2, 3, 11, 12).

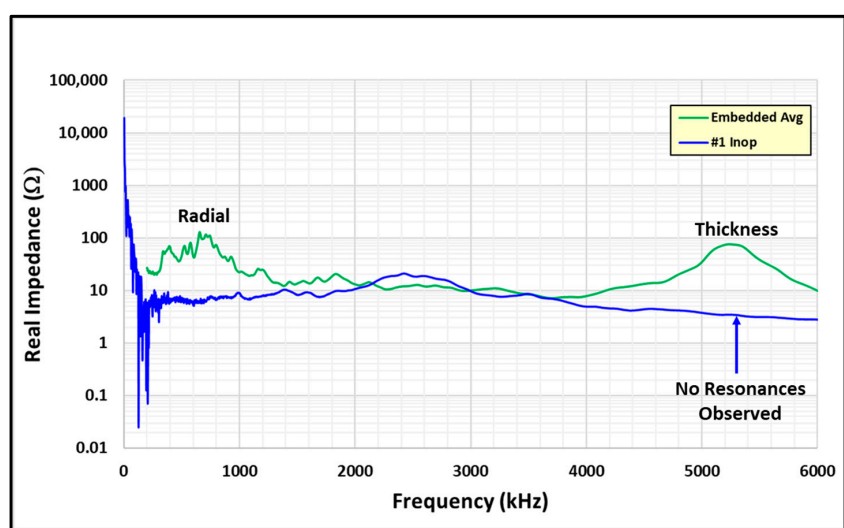

**Figure 14.** Inoperable embedded sensor #1 impedance response.

*3.2. Tensile Load Test Results and Sensor Impedance Behavior*

3.2.1. Pristine Coupon Test Results

The coupons without sensors (5) were tested following the procedure in the previous section with the exception of the second applied load, which showed an increase of 2224 N instead of 445 N. For each additional load, the level was increased by 445 N until failure occurred. The coupon was not removed from the test machine between loads. The average failure load for the pristine coupons was 10,280 N and a bondline stress of 15.9 MPa. For pristine coupon 2, Figure 15 shows the load versus bond displacement on the final load increment to failure with an x on the curve depicting the maximum load achieved on that test. The other pristine coupons exhibited similar load versus displacement plots. The failures were mixed-mode with mostly cohesive failures (in the adhesive). One of the coupons failed in a composite adherend fiber tear.

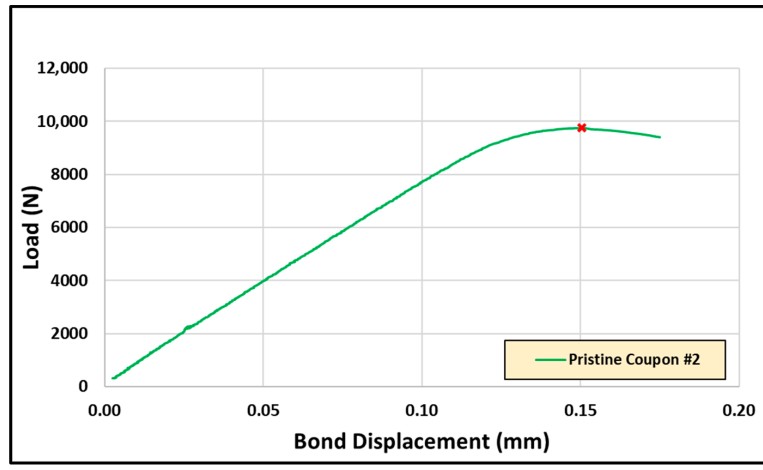

**Figure 15.** Pristine test coupon #2 load versus displacement (final load increment).

3.2.2. Embedded Sensor Coupon Test Results

An example of the applied sequential loadings for an embedded sensor coupon is shown on Figure 16. The sample (with sensor #11) failed at 6630 N on the 7th load increment exhibiting a mixed mode failure (primarily cohesive but some thin layer adhesive). The failed coupon 11 is shown on Figure 17. The cohesive failure mode indicates that failure originated in the bond adhesive, and the thin layer adhesive failure aspect indicates that the failure was close to becoming an adhesive failure at the bond interface. The adhesive

exhibited a high degree of porosity, which was noticeable by the incomplete adherend surface coverage with adhesive. The brittle sensor was fractured into multiple pieces.

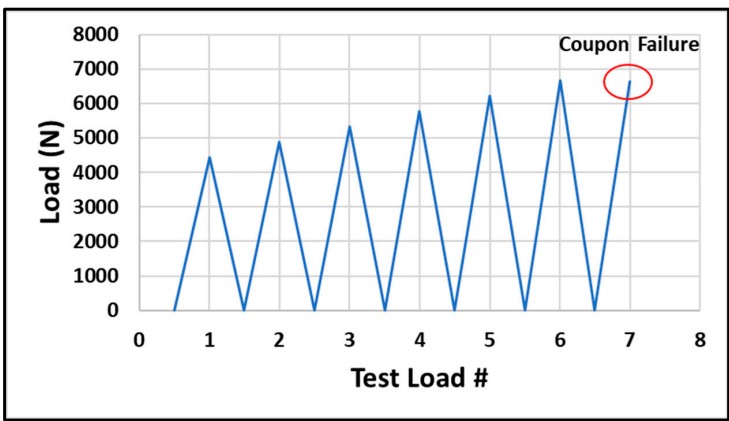

**Figure 16.** Example coupon (11) test loads.

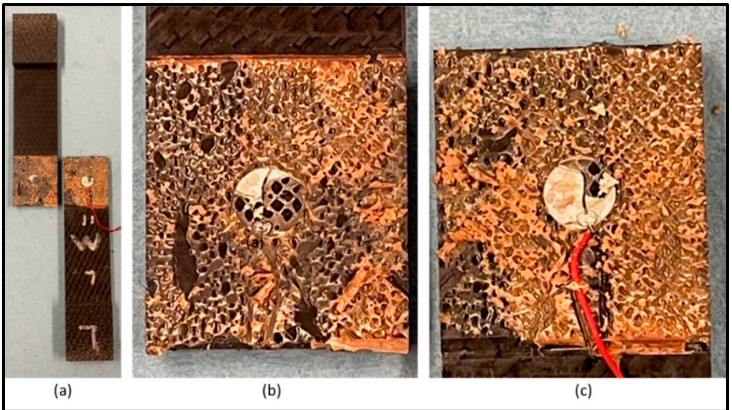

**Figure 17.** Coupon 11 cohesive failure photos with broken sensor disk. (**a**) Both adherends, (**b**) top adherend, (**c**) bottom adherend.

The embedded sensor coupons failed at an average load of 6510 N after six load increments, corresponding to a bondline stress of 10.1 MPa. A plot of the final failure load test versus bond displacement for the tested embedded sensor coupons is shown on Figure 18. The 'x' on each curve indicates the peak load achieved on the coupon.

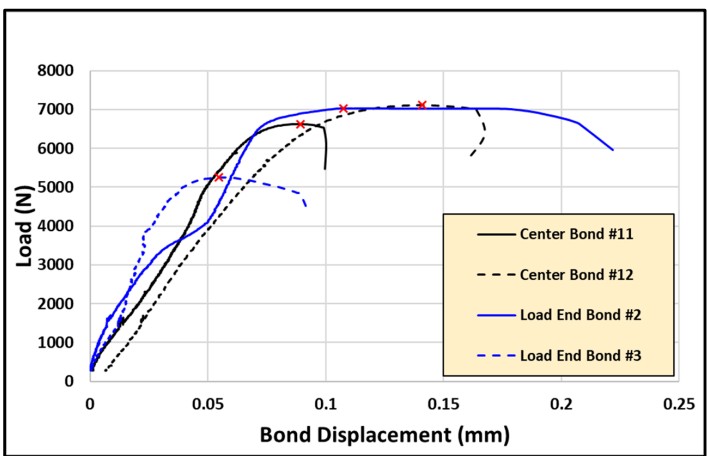

**Figure 18.** Failure load test cycle versus displacement.

Coupon #11 is used to review the results and electromechanical impedance in detail. All coupon failure modes are similar to coupon #11 (primarily cohesive bond failure). The sensor embedded in the center of the lap shear bond achieved the highest load (sensor #12). One of the sensors embedded at the load end of the bond achieved the lowest test load (sensor #3). Table 3 provides the peak load achieved.

**Table 3.** Peak load achieved for each sensor coupon test.

| Sensor | Bond Location | Peak Load (N) | Stress (MPa) |
|--------|---------------|---------------|--------------|
| 2 | Load end | 7033 | 10.9 |
| 3 | Load end | 5257 | 8.1 |
| 11 | Center | 6630 | 10.3 |
| 12 | Center | 7120 | 11.0 |
| Average | | 6510 | 10.1 |

For each sensor coupon, the electromechanical impedance was measured in the unloaded condition between the load increments, repeating up to failure. As an example, Figure 19 shows the real and imaginary values of the impedance for each load level on the sensor #11 coupon. Note that the various colored curves represent each sequential load increment.

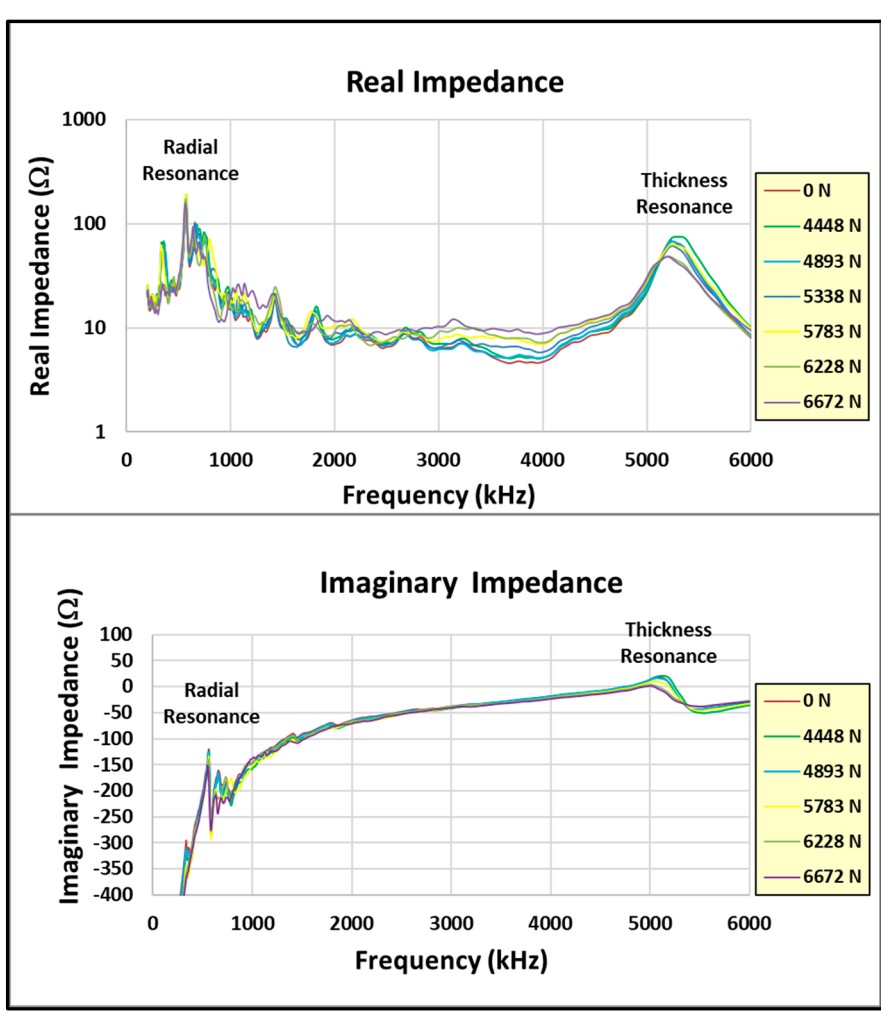

**Figure 19.** Sensor #11 impedance for each load increment.

## 4. Discussion

The discussion that follows is broken down into three subsections related to the sensor itself, the effects of mechanical testing on the embedded sensor specimens, and the effects of sequential loading as a way to review the results and assemble critical outcomes.

### 4.1. Baseline Sensor Impedance Testing

The free state impedance response was first compared to the simulation results, shown in Figure 20. Sensor #11 impedance results were chosen for the comparison. The analytical representation correlates well with the experimental results for the real part of the impedance. This indicates that the simulation captured the structural resonance of the sensor. The small difference in the primary radial resonance is most likely due to the effect of the wires suspending the sensor. This is amplified in the higher-order harmonics. This is not a concern, since embedding the sensor constrains the response and removes any contribution of the higher order resonant responses. The primary thickness impedance showed good correlation to the test results. It is interesting to note that the imaginary impedance showed a much higher amplitude at the radial resonance than the simulation. This is due the free state of the sensor in the experiment and indicates that the sensor was not bonded, which matches the free boundary state of the test. As a result of the acceptable correlation, the ANSYS analytical model can be used directly in a future 3D representation of the joint for embedded sensor impedance simulation studies.

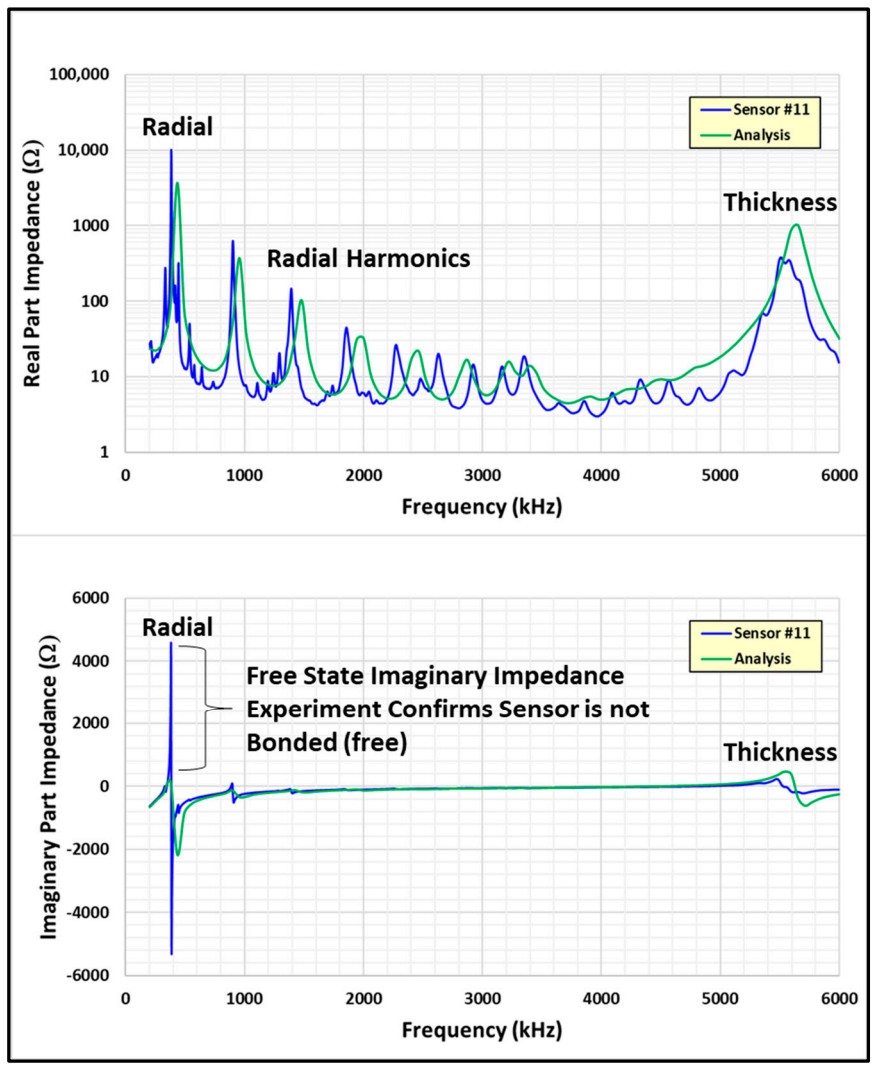

**Figure 20.** Free sensor impedance sensor #11 vs. analysis.

Figure 21 shows the comparison of the baseline impedance tests, both of the sensor in the free state and the embedded joint state before test loading. Sensors 2, 3, 11, and 12 are shown since they are the coupon-embedded operable sensors. The free state response shows a dominant first radial mode, followed by its higher order harmonic resonances. The thickness mode exhibited a clear response at approximately 5500 kHz with lower amplitude and higher damping than the primary radial mode. The embedded coupon showed a much lower amplitude and higher damping for both the radial and thickness modes. This is the physical effect of constraining the sensor within the adhesive bond of the test coupon. The sensor is not allowed to move and exhibits higher damping. Note that the radial harmonic modes are no longer distinct enough to readily separate in the embedded sensor response, as shown in Figure 21b. The resonance responses of the four sensors were averaged and plotted for free versus embedded sensor impedance comparison in Figure 22. The primary radial response increased from an average of 384 kHz to 656 kHz with much lower amplitude and higher damping. The increase in resonant frequency and damping is due to constraining the sensor within the bondline of the coupon, resulting in higher pointwise structural stiffness. The thickness mode response also showed a decreased amplitude with higher damping (broader peak). The thickness resonance decreased in frequency from an average of 5600 kHz to 5200 kHz.

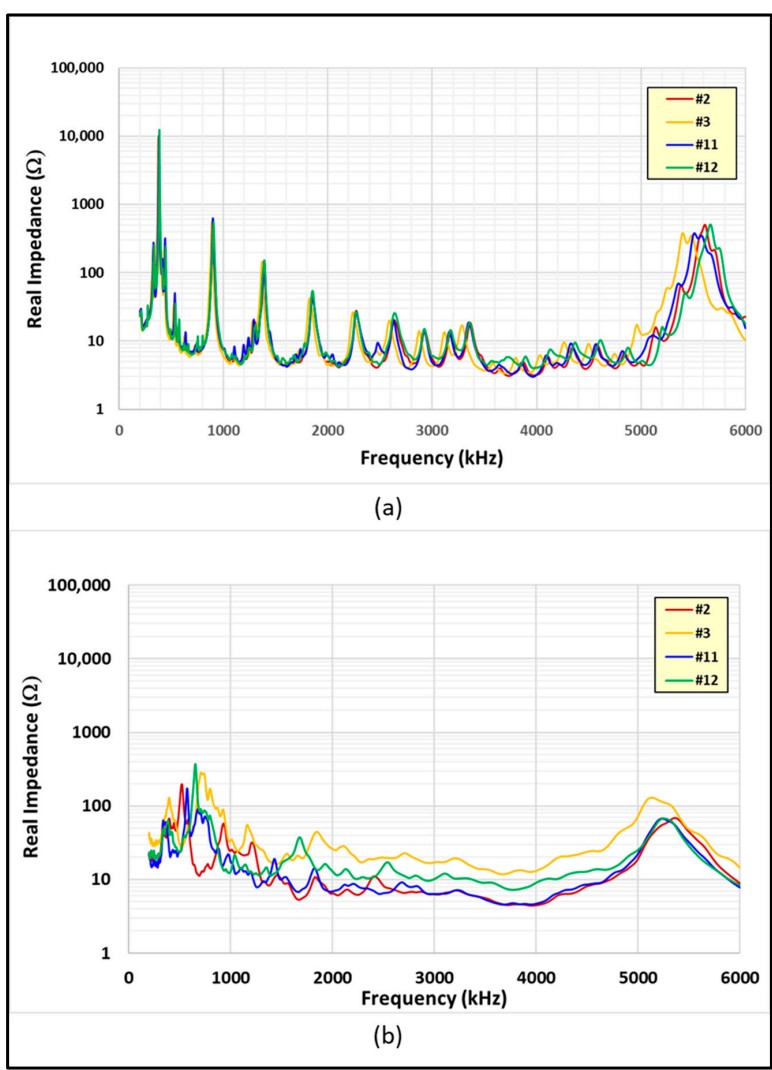

**Figure 21.** (**a**) Free sensor, (**b**) embedded sensor, baseline impedance results.

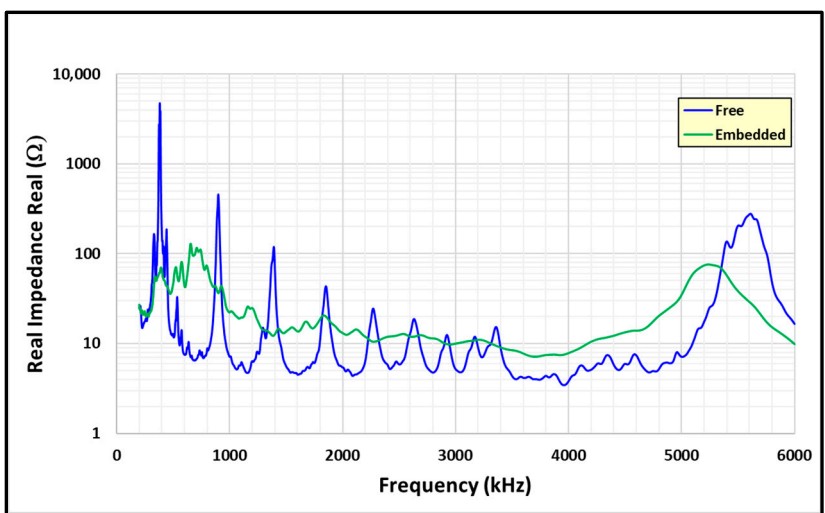

**Figure 22.** Free versus embedded sensor average impedance.

## 4.2. Embedded Sensor Effect on Tensile Load Performance

The lap shear coupon failure loads are summarized in Table 4 and compared to the baseline (without sensor) coupon test results. The load versus displacement for the final load increment (before the failure load test) comparing the three configurations is plotted on Figure 23. Sensor 11 (center bond) and Sensor 3 (load end bond) are plotted with pristine coupon 2 along with the maximum load achieved, designated with an x. Some clip gage slipping is noted in the sensor #3 trace above 2000 N and in the sensor #11 trace above 4000 N. The slopes (stiffnesses) were similar, measured prior to clip gage slipping, for the three configurations, with the embedded sensor coupons failing at lower loads.

**Table 4.** Average tensile failure load.

| Configuration | Failure Load (N) | % Difference |
|---|---|---|
| Pristine | 10280 | – |
| Sensor center | 6875 | −33 |
| Sensor load end | 6145 | −40 |

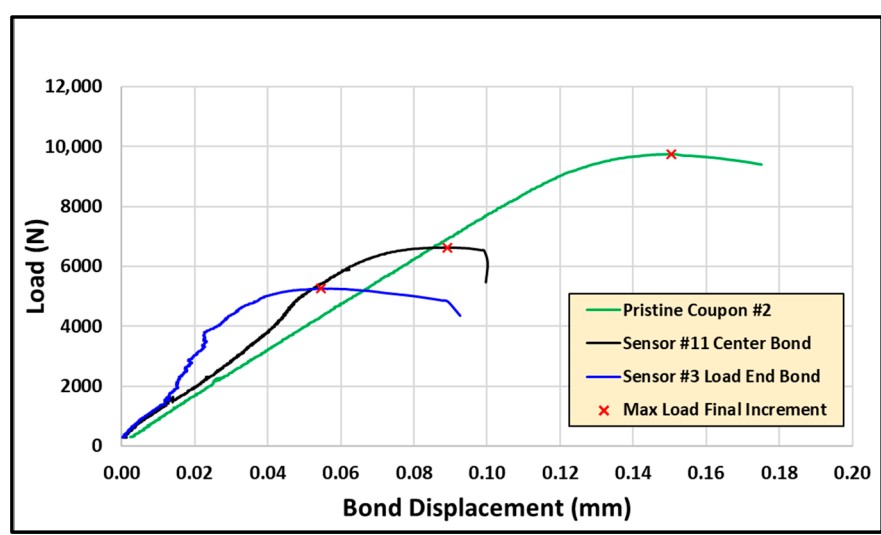

**Figure 23.** Load versus displacement of pristine and sensor coupons.

The failure loads in Table 4 indicate a much lower result when the sensor is embedded in the coupon bondline. The % difference for the sensor located in the center of the bond was −33% and −40% when the sensor was located at the end of the bond. To understand these results, the coupon thicknesses were measured both for the pristine and embedded sensor coupons. The bondline thickness of the pristine coupons measured an average of 0.25 mm, and the sensor coupons were 0.62 mm. The reason for the thicker bondlines of the sensor coupons is due to the sensor and wire thickness values. The sensors measured 0.39 mm thick, which is double the APC specified value (0.20 mm) for this piezoelectric sensor. This was not anticipated and was due to the effect of soldering the wires to the upper and lower surfaces of the sensor. Furthermore, the wires (which are embedded in the bond) had a diameter measured at 0.50 mm each. Together, these items led to a bondline thickness of 0.62 mm, which was measured consistently for both sensor locations. Unfortunately, this thicker bondline (over twice the thickness of the pristine coupons), resulted in adhesive-poor, porous, void-rich bonds for all the sensor coupons. The result of this adhesive-poor bondline is a knockdown of 33–40% on the tensile load performance compared to the pristine specimens. The test data are consistent and represent a bondline that exhibits the effects of these defects.

However, since the embedded sensor coupons were all similar in bondline thickness, meaning all had similar porosity and void content, the failure load results can be compared to determine the effect of sensor location on bond strength. As previously mentioned in the experiments section, a finite element model was constructed and analyzed to simulate the lap shear test and evaluate the sensor effect on the stress distribution through the coupon-adhesive bond.

It is seen in the experimental results that the coupons with the sensors near the endpoints failed at a lower load level than the center bond-located sensors. These results are consistent with the analytical data and indicate that the center of the bond should be selected for the sensor location, based on the structural performance of the lap shear joint.

*4.3. Embedded Sensor Impedance during Successive Loadings*

The radial and thickness resonance results, with increasing load, are plotted in more detail in Figure 24 for the coupon with Sensor 11. Each line is an incremental load cycle that was increased until the coupon failed. A large change was noticed in the thickness response on the final incremental load before failure. This was most likely due to a loss in joint stiffness and a gain in damping, which lowered the resonance amplitude. It is interesting to note that both the radial and thickness resonant frequencies increased slightly from the baseline values after the initial loading. The real impedance amplitude also increased. This could be due to initial consolidation of the joint under a light load. After the initial loading, both resonances began to decrease in frequency and amplitude. A major change was seen in both responses on the final load cycle (highest). The response amplitude was greatly reduced, and the damping was significantly increased. It is also interesting to review the imaginary part of the impedance load response in detail for both resonances. This is plotted in Figure 25.

Of note, in both Figures 24 and 25, examining the thickness resonance change (both real and imaginary) parts reveals a negative change at a lower load level (6228 N) than is shown in the standard evaluation of the radial resonance change. This corresponds to an earlier warning of joint degradation as the loading level was increased.

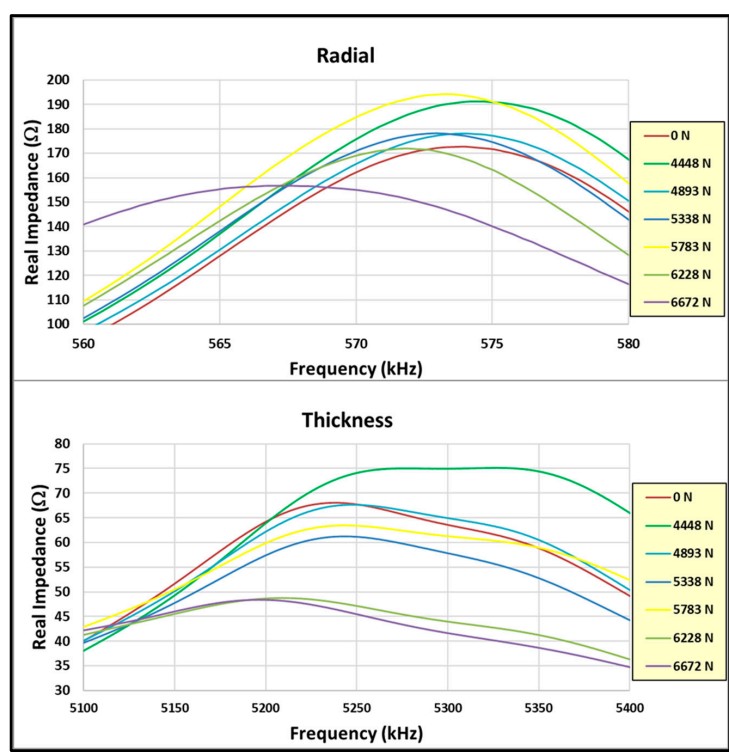

**Figure 24.** Radial and thickness impedance (real) resonance and load level for Sensor 11.

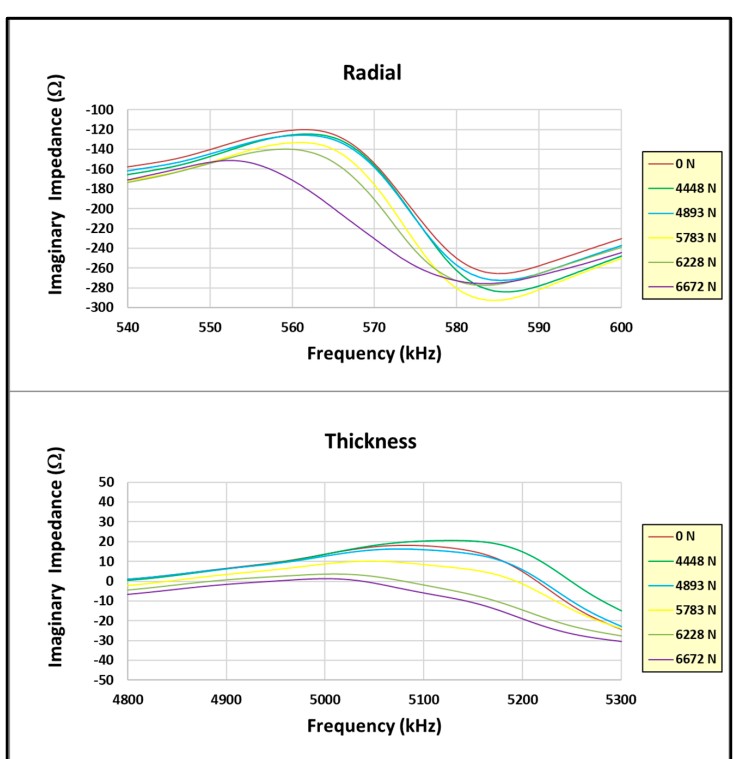

**Figure 25.** Radial and thickness impedance (imaginary) resonance and load level for Sensor 11.

Reviewing the current literature produces several methods to evaluate the structural health of the bond via the impedance response of a piezoelectric transducers [12,30,31]. One of the most prominent evaluation tools is the root mean square deviation (RMSD) of the measured real part of the impedance during successive loadings. The RMSD is used here for the estimate of bond damage. Equation (4) below defines the RMSD, which relates the

most recently measured impedance resonant frequency compared to the baseline measured value. For this work, the damage index was calculated for both of the primary radial and thickness resonances. Most of the literature focuses on the radial mode evaluation of the change in the real part of the impedance. A damage index value of $-2\%$, calculated on the radial resonance, is the estimated threshold where the bondline integrity begins to approach a degradation level that corresponds to joint failure [12].

$$RMSD = \sqrt{\frac{\sum_{i=1}^{n}[Re(Z_n(\omega_i)) - Re(Z_u(\omega_i))]^2}{\sum_{i=1}^{n}[Re(Z_n(\omega_i))]^2}} \qquad (4)$$

where $Z_n$ is the healthy bond impedance and $Z_u$ is the measured bond impedance after the static load is applied and unloaded. Table 5 shows the calculated damage indices for each load level (both radial and thickness resonances) and Figure 26 shows the damage index plot as a function of applied load, again for the embedded Sensor 11 test coupon, with the failure load indicated.

**Table 5.** Load and damage index (DI) sensor #11.

| Load (N) | Radial (kHz) | DI (%) | Thickness (kHz) | DI (%) |
|---|---|---|---|---|
| 0 | 574 | – | 5239 | – |
| 4448 | 574 | −0.1 | 5274 | 0.7 |
| 4893 | 574 | 0.0 | 5248 | 0.2 |
| 5338 | 573 | −0.2 | 5246 | 0.1 |
| 5783 | 573 | −0.1 | 5242 | 0.1 |
| 6228 | 572 | −0.4 | 5208 | −0.6 |
| 6672 | 567 | −1.2 | 5196 | −0.8 |

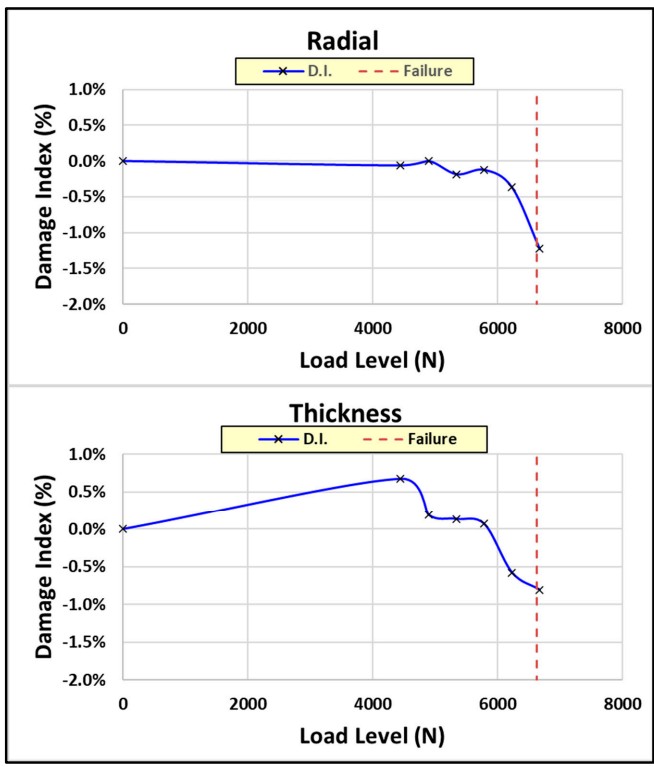

**Figure 26.** Sensor 11 damage index versus load level.

For further definition, Figure 27, shows each measured impedance resonance (radial and thickness), plotted against load level with the failure load denoted and the −2% damage index identified.

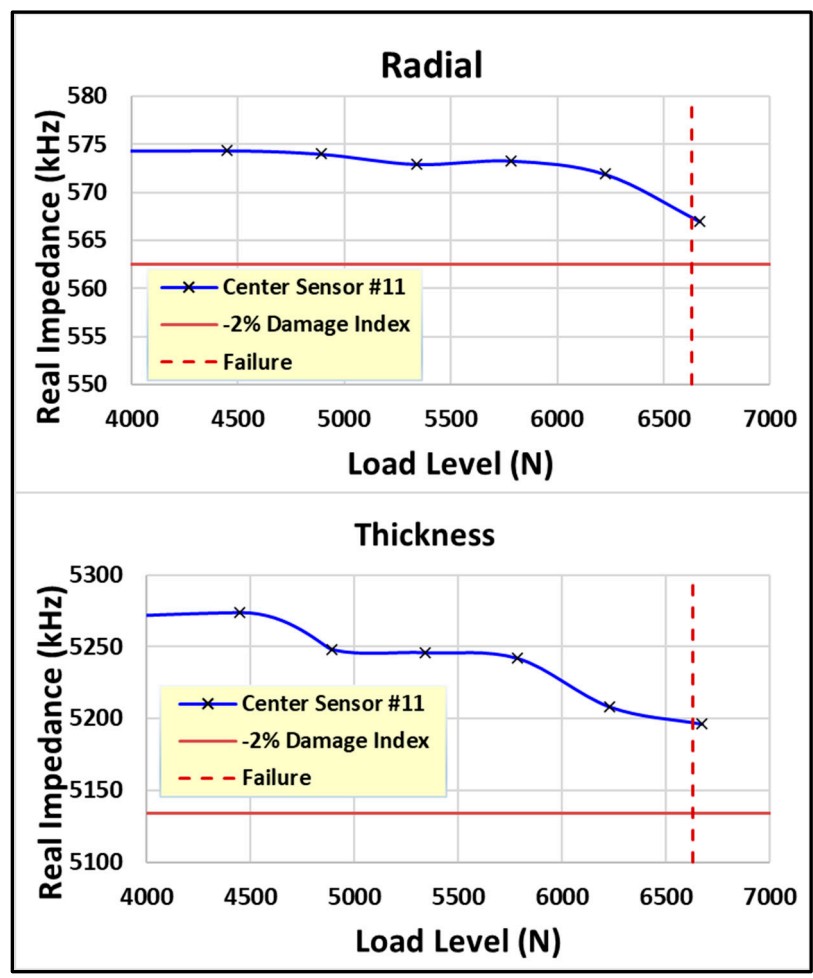

**Figure 27.** Sensor #11 real impedance versus load level.

Both the radial and thickness impedance damage indices trend toward the −2% RMDS value at the time of failure of coupon #11. Traditionally, the damage index is evaluated only for the radial or in-plane response. For this work, the thickness, or out-of-plane response, was also evaluated with repeated loads. The radial response dropped off quicker than the thickness response and was close to −2% at failure. However, evaluating the thickness impedance damage index shows a larger change at a lower load level than exhibited in the radial damage index load progression. This is illustrated in Figures 26 and 27. From these plots, it is seen that the change in sensor impedance response can be used to warn of impending joint failure, and that examination of the thickness resonance impedance change can provide an earlier alert to degradation of the joint. Of particular note, on the second-to-last load before failure, 6228 N, a significant change in the damage index is observed for the thickness impedance response suggesting that failure is imminent and structural loading should be reduced to prevent catastrophic failure.

The test coupons with the sensors located at the load end of the bond behaved differently than the center bond sensors in terms of the change in impedance resonance or damage index. These coupons exhibited a large damage index after the first load was applied. For both load-end bond sensors, the radial impedance damage indices were greater than the −2% warning level after this loading (4448 N). However, both of these coupons continued to take load and failed at much higher levels than the −2% warning

level. Load-end bond sensor #2 failed at 7033 N and sensor #3 failed at 5257 N. Locating the sensors at the end of the bond in the area of high stress has more effect on the impedance response of the sensor during loading and aids in earlier failure initiation. This adds difficulty to using the damage index to predict joint failure.

Due to the degradation in structural performance with the sensor location at the end of the bond and the difficulty observed with the damage index failure load predictability, it is recommended to use the center of the bond for locating the sensor. This location minimizes the degradation on tensile load shear performance and produces impedance results, during repetitive loadings, that can be used for bond health monitoring.

## 5. Future Work

For future work, we plan to fabricate test coupons using additional layers of film adhesive to a create a porous/void-free bond. These coupons would have the same sensor (thickness and attached wires). Sequential loadings in both the tensile and normal (flatwise tension) directions are then to be performed to evaluate the prognostic capability using both the radial and thickness impedance resonances. Single lap shear fatigue loads (cyclic) are also planned to evaluate the effect of repeated operational loads on the structural integrity of the joint. These planned adjustments are expected to more closely approximate a real-world aircraft composite joint and allow for studies to be performed with a 3D-coupled field analysis model.

## 6. Conclusions

An approach was developed, based on using bondline-embedded piezoelectric disk sensors, to evaluate the integrity of the bond at fabrication and during operational usage. Experiments were performed on state-of-the-art composite single lap shear joints, with and without sensors embedded within the joint, and incrementally tensile-loaded until failure. The damage accumulation of the joint was successfully tracked through the evolving impedance of the embedded sensor, showing a substantial change in the incrementally increasing loads preceding failure. A novel approach for health monitoring, with piezoelectric sensors, was demonstrated by calculating and evaluating the damage index based on the higher frequency thickness resonance. A review of the thickness resonance change led to an earlier notification of joint degradation than when reviewing the radial resonance change by itself. Embedding the sensor in the center of the bond demonstrated a smaller reduction in failure load than the sensor located near the load end as compared to the pristine coupon results. This was due to the center location in the shear joint having the lowest maximum adhesive shear stress in the joint and is consistent with the analytical prediction. The sensors located in the center of the bond were observed to also better predict the shear joint failure than the sensors located near the end of the bond where the stress is concentrated. The load-end bond sensors exhibited more impedance change during loading, with the changes starting at lower load levels; however, locating the sensor in the stress concentration area tended to initiate joint failure at lower load levels. The outcomes of the experiments indicate that piezoelectric sensors embedded in the bondline of composite single lap shear joints can alert impending joint failure through measurement of the joint electromechanical impedance change of both the radial and thickness resonances during repeated loadings and are best located at the center of the bond where the lowest joint stress levels exist.

**Author Contributions:** S.P.C. and D.W.R. were involved in conceptualization, methodology, and writing the original draft; S.P.C. was involved in investigation and formal analysis; D.W.R. was involved in the writing's review and editing, project administration, and funding acquisition. All authors have read and agreed to the published version of the manuscript.

**Funding:** No external funding was utilized in the completion of this research.

**Data Availability Statement:** The data presented in this study are available upon request from the corresponding author.

**Conflicts of Interest:** The authors declare no conflict of interest.

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
