# Peer review of "Composite Single Lap Shear Joint Integrity Monitoring via Embedded Electromechanical Impedance Sensors"

_jcs, doi:10.3390/jcs7020053_

Round 1
Reviewer 1 Report
In the article authors study an approach, based upon using bond embedded piezoelectric disk sen-sors, to evaluate the integrity of the bond at fabrication and during operational usage. This study is undoubtedly of great practical interest. Therefore, this article has the potential for publication. However, from a scientific point of view, the article looks insufficiently developed and needs to be improved.
1. It is desirable to indicate what material the piezoelectric sensor is made of. What are the main parameters of the sensor (please give numbers)? It makes sense to describe the experimental technique in more detail.
2. The impedance characteristics of sensors are of interest in both the real and imaginary parts. How does the environment, loading and constraines affect sensor performance?
3. Imaginary impedance can not be measured in degrees (see Fig 15). Please clarify.
4. Why do the authors neglect the imaginary part of the impedance in their studies, excluding only one graph in Figure 15? Analysis of both the real and imaginary parts (Nyquist diagrams, for example), would not give more information about the object being measured?
5. This article looks too applied. I would like to see more material and measurements analysis, model representations, physical interpretation.
6. What is the scientific novelty of the work? Please put more emphasis on this.
Author Response
Reviewer #1 Comments/Author Actions
- It is desirable to indicate what material the piezoelectric sensor is made of. What are the main parameters of the sensor (please give numbers)? It makes sense to describe the experimental technique in more detail.
Response: The sensor material details were added to the experimental section, including mechanical and electrical properties that were used in an analytical simulation for the sensor free state response. More details were added to the experimental technique in the introduction and experimentation sections. (See: page 3, Section 2.1)
- The impedance characteristics of sensors are of interest in both the real and imaginary parts. How does the environment, loading and constraints affect sensor performance?
Response: More information was added to the details of the complex impedance quantity, with a detailed discussion on the characterization of the real part and the imaginary part. The real part is directly related to the pointwise mechanical impedance, which is the same as the structural resonance of the joint. The imaginary part is much less sensitive to the structural resonance and is applicable to the integrity of the sensor itself. A change in the imaginary impedance could indicate a local disbond of the sensor but not the structural state of the joint. The effect of the loading was addressed by the added simulation and used to inform the location of the sensors for testing. The load test results confirmed the simulation. The effects of the environment and test coupon constraints were not addressed. (See: page 3, sentence 8)
- Imaginary impedance cannot be measured in degrees (see Fig 15). Please clarify.
Response: Figure 15 was mistakenly labeled, although the graph was correct. The label was corrected replacing degrees with Ohms. (See: page 16, Revised figure #18)
- Why do the authors neglect the imaginary part of the impedance in their studies, excluding only one graph in Figure 15? Analysis of both the real and imaginary parts (Nyquist diagrams, for example), would not give more information about the object being measured?
Response: The real part of the impedance is the primary focus for the structural health of the joint; however, the comment is understood, and the imaginary part has been added to some the embedded sensor plots. (See: page 12, Revised figure #12,18, 19, 25)
- This article looks too applied. I would like to see more material and measurements analysis, model representations, physical interpretation.
Response: Sensor material information was added to the article. A sensor free state simulation was added. (See: pages 4 & 5) A static analysis model was added to show the derivation of the embedded sensor location on the bond shear stress distribution. (See: page , figure 5 )
- What is the scientific novelty of the work? Please put more emphasis on this.
Response: Emphasis was added to show the scientific novelty of including the evaluation of the thickness impedance resonance change in addition to the standard evaluation of the radial resonance in the RMSD damage index. (See: page 3, last sentence of Section 1 and the 1st paragraph on page 23 )?
Reviewer 2 Report
The experimental portion was carried out correctly. English proficiency is quite good. Science-wise, the article's subject matter is pertinent. I thus think that this work's concept is within the purview of the Journal of Composites Science. However, There are some particular aspects that need to be addressed before publication, therefore I recommend major revision of the following points:
1. Many things like ‘Test Procedure’ on Page 7 should be written in proper format and should be included in Experimentation.
2. If possible, finite element model (FEM) should be developed to simulate the EMI behavior of the embedded sensors to obtain a qualitative representative result of the impedance behavior over the large frequency bandwidth.
3. Comparison of numerically simulated and experimentally obtained impedance behavior should be made.
4. Impedance Behavior should be checked using cyclic external loads of Embedded Piezoelectric Sensors after Fatigue Loading.
5. Recent references should be included in the manuscript.
6. Please include error bars in Figure 20-22
7. Please use contrasting colors in Fig 9 for #2 and # 11 to make difference more visible
Author Response
Reviewer #2 Comments/Author Actions
- Many things like ‘Test Procedure’ on Page 7 should be written in proper format and should be included in Experimentation.
Response: The ‘Test Procedure’ was put in proper format and included in the Experimentation section. (See: page 11, Table 2)
- If possible, finite element model (FEM) should be developed to simulate the EMI behavior of the embedded sensors to obtain a qualitative representative result of the impedance behavior over the large frequency bandwidth.
Response: An Ansys FEM was included to simulate the EMI behavior of the sensor in the free state boundary condition. The authors are currently working on an embedded sensor simulation that is planned for future publication. (See: pages 4 & 5)
- Comparison of numerically simulated and experimentally obtained impedance behavior should be made.
Response: Comparison of the simulated and experimental impedance behavior was added. (See: figure 19)
- Impedance Behavior should be checked using cyclic external loads of Embedded Piezoelectric Sensors after Fatigue Loading.
Response: Cyclic fatigue loading was not performed as part of this studies experiments. The sensor coupons were loaded statically in stepped loads which were increased until the coupon failed. The manuscript wording referring to loading was updated to reflect incremental loads instead of cyclic loading. The impedance is measured in the unloaded state after the load is applied and removed.
- Recent references should be included in the manuscript.
Response: After a literature review, most of the updated works referred to current status of structural health monitoring. (Reference 15 was added)
- Please include error bars in Figure 20-22.
Response: These figures may not have been clear. They represent single tests on the sensor #11 test coupon. Figure 20 shows the radial and thickness impedance resonance for sequentially increasing loads on the sensor #11 coupon. The x represents the maximum response for each load, which is now removed for clarity. Figure 21 plots the damage index against load level with the x’s representing the data points from the sensor #11 test. Figure 22 plots the same data but has the impedance on the y axis instead of the damage index.
- Please use contrasting colors in Fig 9 for #2 and # 11 to make difference more visible.
Response: Figure 9 has been corrected with contrasting colors for #2 and #11.
Round 2
Reviewer 1 Report
Thanks to the authors for the work done on the corrections, the improvements have been satisfactorily completed. The experiment was carried out correctly, the subject of the article corresponds to the journal. I believe that the article can be published after minor revisions (I would recommend to arrange the results and especially trivial parts more compactly and also make the design of figures more uniform and visual).
Reviewer 2 Report
Authors have done a good job in improving the manuscript. Authors conclusions are justified given the data.